# Serum Amyloid A as a Potential Biomarker in Inflammatory Bowel Diseases, Especially in Patients with Low C-Reactive Protein

**DOI:** 10.3390/ijms25021177

**Published:** 2024-01-18

**Authors:** Marie Stute, Martin Kreysing, Markus Zorn, Patrick Michl, Annika Gauss

**Affiliations:** 1Department of Gastroenterology and Hepatology, Heidelberg University, University Hospital, INF 410, 69120 Heidelberg, Germany; et224@stud.uni-heidelberg.de (M.S.); martin.kreysing@med.uni-heidelberg.de (M.K.); patrick.michl@med.uni-heidelberg.de (P.M.); 2Central Laboratory of University Hospital Heidelberg, Department of Endocrinology and Metabolism, University Hospital Heidelberg, INF 671, 69120 Heidelberg, Germany; markus.zorn@med.uni-heidelberg.de

**Keywords:** Serum Amyloid A, SAA, inflammatory bowel disease, biomarker, Crohn’s disease, ulcerative colitis

## Abstract

The acute phase protein Serum Amyloid A (SAA) is synthesised by the liver in response to inflammatory stimuli. Previous studies have revealed that SAA may be a better biomarker of disease activity in inflammatory bowel disease (IBD) compared to C-reactive protein (CRP). This retrospective monocentric study evaluated whether SAA correlates with biomarkers like faecal calprotectin (FC), CRP, the Neutrophil to Lymphocyte ratio (NLR), the platelet count and clinical disease activity of IBD patients. Serum samples from the IBD outpatient clinic of the University Hospital Heidelberg were analysed for SAA concentrations if an FC concentration measurement was available from ±14 days to collection of the serum sample. Three hundred and six serum samples from 265 patients (166 with Crohn’s disease, 91 with ulcerative colitis and 8 with IBD unclassified) met the inclusion criteria. There was a significant positive correlation between SAA and FC, CRP, NLR, platelet count and the Simple Clinical Colitis Activity Index (SCCAI). The cut-off for SAA serum concentration at 4.55 mg/L achieved a sensitivity of 57.5% and a specificity of 69.7% for the detection of active inflammation in IBD. SAA may be used as an additional biomarker in the disease monitoring strategy of IBD patients, especially in patients with low CRP concentrations.

## 1. Introduction

The term inflammatory bowel disease (IBD) includes the two disease entities Crohn’s disease (CD) and ulcerative colitis (UC), as well as IBD unclassified (IBDU). All of them are chronic intestinal inflammatory conditions characterised by phases of remission and active disease. However, during a phase of clinical remission, there are often still signs of active inflammation detectable via endoscopy [1].

It has been shown that the achievement of endoscopic remission is associated with more favourable long-term outcomes, such as a reduced need for corticosteroid therapy and hospitalisations [2].

To detect phases of active mucosal inflammation, regular surveillance of disease activity is essential. Sometimes a new disease flare has already started before the patient has recognised any clinical symptoms [3,4]. So far, the most common tool used to track down mucosal inflammation in the gut is endoscopy in combination with observed clinical symptoms and blood or stool biomarkers. However, there are some disadvantages that should be taken into consideration. For instance, endoscopy is invasive, time-consuming and, above all, a substantial burden to the patient [5].

Therefore, many studies are investigating non-invasive biomarkers for the surveillance of disease activity in both CD and UC. At present, the most common biomarkers used in this context are C-reactive protein (CRP) and faecal calprotectin (FC) [3]. Nevertheless, they have limitations in terms of their use and reliability. For FC, the collection of stool samples and pre-analytics are identified as potential pitfalls. Many patients would prefer a blood-based determination of biomarkers [6]. CRP is not specific to the bowel and may therefore be increased due to inflammatory conditions elsewhere in the body. While CRP may be elevated due to IBD activity, the cause of its elevation cannot be isolated to one single site of inflammation [7].

Due to the lack of a blood biomarker which can represent the activity of IBD with high specificity and sensitivity, a few studies have explored the use of Serum Amyloid A (SAA). SAA is an acute phase protein (APP) bound to high-density lipoproteins (HDL). Due to its induction via proinflammatory cytokines, SAA levels can increase 1000-fold during phases of acute inflammation [8]. In studies on rheumatoid arthritis or chronic active hepatitis, SAA proved to be a more sensitive marker to assess active inflammation as compared to CRP. Previous studies on SAA in IBD found a correlation between SAA concentrations and endoscopic activity as well as different known biomarkers [1].

The aim of this study was to explore if SAA correlates with FC and other biomarkers of inflammation in CD and UC patients, and whether it may be used as an additional inflammatory marker to detect subclinical inflammation.

## 2. Results

### 2.1. Patient Characteristics

Table 1 summarises the characteristics of the included patients. In total, 306 serum samples from 265 individual patients were analysed for SAA concentrations. The median age of the patients was 44 years (range: 18–77 years). One hundred and eighty-two serum samples were from CD patients, 116 from UC patients and eight from patients with IBDU. One hundred and fifty-nine serum samples were from female patients and 147 from male patients. For 22 patients, more than one serum sample was available. On average, the included patients were diagnosed with IBD 10 years (range: 0–47 years) prior to the serum sample collection. About 50% of the patients received biologic or small molecule therapy for their IBD (including infliximab, adalimumab, golimumab, vedolizumab, ustekinumab, tofacitinib and calcineurin inhibitors). The mean time interval between the date of FC determination and serum sample acquisition for SAA determination was 3.7 (± 4.4) days.

The laboratory markers of patients with arthritis (peripheral and axial) were compared to the rest of the cases to detect potential differences. In the group with arthritis, the median SAA concentration was 15.6 mg/L compared to 20.0 mg/L for the cases without arthritis. The Mann–Whitney U Test was carried out for all laboratory markers and only showed a significant result for the CRP concentration (*p* = 0.031). Patients with arthritis had a median CRP concentration of 2.60 mg/L compared to 1.50 mg/L for patients without arthritis.

Table 2 shows the distribution of the laboratory markers. The median SAA concentration was 4.70 mg/L (IQR: 7.50) for all samples. For CD patients, the median SAA concentration was 4.20 mg/L (IQR: 6.80), for UC patients, it was 5.25 mg/L (IQR: 7.97) and for IBDU patients, it was 5.40 mg/L (IQR: 11.9). The total number of patients with FC concentrations > 1800 µg/g was 29.

There was no significant correlation between gender and disease entity (*p* = 0.130), nor gender and presence of active inflammation (FC > 50 µg/g) (*p* = 0.523). The relationship between disease entity and the presence of inflammation was also not significant (*p* = 0.726). Comparing category L (disease location) for CD, a significant difference between patients with category L1 (terminal ileum) compared to category L2 (colon) and L3 (ileocolon) was found regarding the presence of active inflammation (FC > 50 µg/g) (*p* = 0.005, Cramer’s V = 0.216). Similar results were calculated for UC. After performing the Chi^2^ test for category E in UC patients, there was a significant difference between E1 (*p* = 0.021, Cramer’s V = 0.215) and E3 (*p* = 0.003, Cramer’s V = 0.275).

The Mann–Whitney U Test revealed that the difference in SAA concentrations between the subgroup with active inflammation (FC > 50 µg/g) (median: 5.20 mg/L) and without active inflammation (FC < 50 µg/g) (median: 3.45 mg/L) was significant (*p* < 0.001, r-value = 0.217). Similarly, the results indicated a significantly larger difference in SAA concentrations when comparing the patients based on their CRP concentration (*p* < 0.001, r-value = 0.558). As a secondary finding, the test showed a significant difference in the median NLR for patients who were on systemic steroids (median: 6.59) as compared to those without systemic steroid therapy (median: 2.29) (*p* < 0.001, r-value = 0.448).

### 2.2. Correlation Analyses

There was a positive correlation between SAA and FC concentrations (r = 0.365, *p* < 0.001) (Figure 1), SAA and CRP concentrations (r = 0.623, *p* < 0.001) (Figure 2), SAA concentrations and NLRs (r = 0.381, *p* < 0.001) (Figure 3), SAA concentrations and platelet counts (r = 0.375 *p* < 0.001) (Figure 4) and SAA concentrations and SCCAI scores (r = 0.300, *p* = 0.001). The correlation between SAA concentrations and the HBI in CD patients was not significant (r = 0.082, *p* = 0.276). The difference in the coefficient of correlation between SAA and FC concentrations for the various disease entities was only small (CD = 0.376, *p* < 0.001; UC = 0.300, *p* = 0.001), and the correlation was not significant for IBDU (r = 0.700, *p* = 0.053). Comparing all groups, the strongest correlation was consistently found between SAA and CRP concentrations. Regarding the correlations between SAA concentrations and NLRs, the correlation coefficient was higher for patients with CD (r = 0.405, *p* < 0.001) than for those with UC (r = 0.294, *p* = 0.002). After dividing the cases with CD into subgroups according to disease location, the group L1 (terminal ileum) showed the lowest correlation for SAA with all the analysed biomarkers. In the same way, patients with UC were categorised based on disease extent (Montreal E). The subgroup E3 (extensive colitis) was the only one with a significant positive correlation between SAA and FC concentrations (r = 0.424, *p* < 0.001). In addition, all cases were divided into two subgroups according to the intake of systemic steroids in order to compare the coefficient of correlation for SAA and the NLR. It was significant for the cases without systemic steroid intake (*n* = 249, r = 0.294 *p* < 0.001) and non-significant for those on systemic steroids (*n* = 54, r = 0.205 *p* = 0.138). Finally, the correlation coefficient for CRP and FC concentrations was compared with that for SAA and FC concentrations. They differed only slightly (SAA and FC: r = 0.381, *p* < 0.001; CRP and FC: r = 0.365, *p* < 0.001) (Figure 5). Further coefficients of correlation can be found in Appendix A.

### 2.3. Linear Regression Analyses

To quantify the relationship between SAA and FC concentrations, a linear regression analysis was performed. The logarithmic values of both variables were used to confirm normal distribution. A positive correlation was shown with an increase of 0.125% in SAA when the value for FC increased by 1%. In the same way, SAA increased by 0.327% when CRP increased by 1%. In comparison to this, the change in CRP concentration was 0.121% when the FC concentration increased by 1%.

### 2.4. Cut-Off Value for SAA Concentration

To define active inflammation, FC was used with a cut-off value of >50 µg/g. This is the upper limit of normal for the assay that was used by the laboratory. The optimal cut-off value for SAA after ROC analysis to detect active inflammation was 4.55 mg/L. SAA achieved a sensitivity of 57.9% and a specificity of 69.7% for the detection of active inflammation. In comparison, CRP reached a lower sensitivity of 34.8%, but a higher specificity of 83.3% using the given cut-off value of 5 mg/L. For this study, the optimal cut-off for CRP concentration was calculated to be 2.95 mg/L using the Youden Index. It achieved a sensitivity of 48.3% and a specificity of 78.8% for the detection of active inflammation. In addition, the AUC was calculated. For SAA, the AUC was 0.649 (Figure 6), and for CRP, it was 0.631 (Figure 7). According to these calculations, the number of cases with normal SAA concentrations and FC > 50µg/g was 102.

The optimal cut-off value for SAA concentration differed between patients with CD and those with UC according to the calculations using the Youden Index. While the same cut-off value of 4.55 mg/L with a sensitivity of 54.2% and a specificity of 79.9% was calculated for CD patients, the optimal cut-off value for UC patients was 8.05 mg/L. It achieved a sensitivity of 43.8% and a specificity of 81.5% in detecting the state of active inflammation.

### 2.5. Patients with CRP < 5 mg/L

Figure 8 shows the relationship between SAA and FC for patients with CRP <5mg/L. Using FC > 50 µg/g as an indicator of active inflammation, the subgroup of patients with normal CRP concentrations (<5 mg/L) (*n* = 212) included 157 patients with active inflammation. Seventy (44.6%) of them showed increased SAA concentrations (>4.55 mg/L) despite displaying normal CRP concentrations. In contrast, only 15 patients with normal SAA concentrations (<4.55 mg/L) and elevated FC concentrations had increased CRP values (>5 mg/L). This accounts for only 14.7% of the group with active inflammation.

## 3. Discussion

This study confirms a positive relationship between SAA concentrations and concentrations of established inflammatory biomarkers in IBD. In current recommendations where treat-to-target strategies for IBD patients were evaluated, the normalization of faecal and serum biomarkers was identified as a medium-term treatment goal [9]. Although clinical remission and endoscopic healing are two of the most important long-term targets in the treatment of IBD patients, their assessment is challenging due to unreliable scores for clinical activity and the risks, costs and burdens that are associated with frequent endoscopic examinations [9]. Their simple access, non-invasiveness and good correlation with endoscopic results build a strong case for the usefulness of biomarkers in the monitoring strategy of IBD patients [9]. While patients would define symptomatic relief as the most important treatment goal, the discrepancy between symptoms and the state of endoscopic disease activity is a common problem. This challenge occurs especially in patients with CD [3,10]. In this context, the use of biomarkers to estimate the state of inflammation is important and supports the clinician in evaluating the disease activity of the individual patient. While endoscopy still provides a more detailed picture of the actual state of inflammation in the bowel, the use of biomarkers such as CRP or FC saves time and reduces the costs of the examination [9]. For these reasons, it is recommended to perform biomarker testing more frequently to avoid periodic endoscopy [11].

Although there have been numerous investigations on the use of CRP and FC in patients with IBD, there is still a need for more sensitive and specific biomarkers to reliably assess disease activity. Hence, this study evaluated the potential of SAA as a biomarker regarding disease monitoring strategies. SAA production increases during the acute phase reaction (APR) and its concentration can be easily determined in a serum sample [12]. Previous studies already found a correlation between SAA concentrations and inflammatory markers such as CRP, FC and interleukins, as well as clinical disease activity and endoscopic findings [1,8,13]. Nevertheless, other studies in this context could not confirm a significant correlation between some of these parameters and SAA. For instance, Bourgonje et al. did not find a significant correlation between SAA and FC, or SAA and the HBI [14].

In this study, SAA concentrations were positively correlated with CRP and FC concentrations, which are currently used as routine biomarkers for the assessment of disease activity in IBD patients. Moreover, there was a positive correlation between SAA concentrations and NLRs, platelet counts and the SCCAI, which is an indicator for clinical disease activity in patients with UC. As mentioned previously, prior studies revealed that indices of clinical disease activity in patients with CD poorly correlate with biomarkers or endoscopic activity indices [14]. Similarly, in this study, the correlation between HBI scores and SAA concentrations in CD patients was not significant. This is one of the reasons why it is recommended to use clinical indices in combination with biomarkers and to avoid relying solely on clinical disease activity scores when making treatment decisions for IBD patients [15]. While the correlation between SAA und CRP concentrations was strong—as demonstrated in previous studies—the correlation between SAA and FC concentrations was only medium [8,10]. In line with our results, the correlation coefficients between SAA and FC concentrations in prior studies were also of medium strength [13,16]. In addition, there was a disparity in the strength of the correlation between cases with isolated ileal disease and those with colonic or ileocolonic disease. The relationship between SAA and FC concentrations was weaker in patients who had isolated ileal disease exclusively. Accordingly, in previous studies evaluating the correlation of FC concentrations and endoscopic disease activity, FC concentrations also showed a stronger correlation in CD patients without isolated ileal disease [6,17].

To our knowledge, there have not been any investigations on the correlation between SAA concentration and the NLR. Our study revealed a moderate correlation between NLRs and SAA concentrations, and a weak correlation between NLRs and CRP and FC concentrations. The latter was already discovered in previous studies [18]. The relationship between SAA concentration and the NLR was stronger for patients with CD compared to patients with UC. Some of the patients were on corticosteroid therapy at the time of sample acquisition. The intake of systemic corticosteroids may increase the NLR and therefore distort the results [19]. This is an important limitation because it may affect the evaluation of the correlation coefficient of SAA and the NLR in this group of patients. In the analysed cases, the subgroup of patients who took corticosteroids had a higher NLR as well as higher FC values than the subgroup of patients without corticosteroid therapy. While this suggested that the state of inflammation may explain the significant positive correlation between SAA concentration and the NLR, the correlation coefficient for the group of patients under systemic steroid therapy was not significant. Therefore, it cannot be ruled out that the influence of the systemic steroids on the NLR outweighs the influence of inflammatory mediators in this specific group of patients.

Platelet count is another laboratory parameter which can be used to detect disease activity in IBD patients. Platelets play a role in the activation of the immune response and in the process of immunothrombosis as part of the intravascular immunity [20]. As IBD patients show abnormalities of platelet numbers and function, their determination and correlation with SAA was also included in this study [21]. The positive and significant relationship between platelet counts and SAA in this study may be linked to their increased levels in the inflammatory immune response, which also plays a role in the disease activity of CD and UC patients [20,22].

CRP is one of the most frequently used biomarkers to detect inflammatory activity in the body, not only in IBD patients but also in patients with other inflammatory diseases. One disadvantage of CRP as a biomarker is the low sensitivity for intestinal diseases due to its increase during any state of systemic inflammation [7]. This creates ambiguity as to whether the CRP value is high because of the activity of the IBD or because of an additional inflammatory focus. Thus, it can lead to discrepancies in the evaluation of disease activity and, potentially, an unnecessary adjustment of the patient’s treatment. The production of SAA is also induced during various inflammatory processes. Consequently, the significance of its changes in concentration is limited due to its function as an APP [12]. Yet, SAA also shows a local increase of production at the site of inflammation, whereas local CRP production only accounts for a small percentage of the serum levels [12,23].

Compared to the serum markers SAA and CRP, FC is more sensitive for intestinal inflammation [15]. Regardless of its higher sensitivity, an important limitation is that there is yet no optimal cut-off for FC to define the state of active disease in patients with IBD [3]. Although this is a considerable problem and its values can vary to a high degree both from day to day and during the course of one day [6], it still outperforms CRP as a marker of inflammation in patients with IBD [9]. Accordingly, the FC value was used in this study as a reference for the state of disease activity and to determine an optimal cut-off for SAA concentration to detect active inflammation. An SAA value of 4.55 mg/L was determined as the optimal cut-off with a sensitivity of 57.9% and a specificity of 67.9%. In comparison to this, previous studies found higher cut-off values in combination with higher values for the sensitivity and specificity of SAA. They also had a higher AUC in predicting endoscopic inflammation [1,8,10]. The accuracy of their analyses may have been improved by the use of endoscopic scores instead of biomarkers.

In addition to the findings above, SAA showed a higher sensitivity than CRP in predicting active inflammation. SAA was able to identify nearly half of the patients with a high FC concentration despite having normal CRP values. This shows the possible strength of the determination of SAA in assessing mucosal inflammation. Due to genetic variations, some patients are missing a CRP response to an acute inflammatory stimulus [24]. This issue has also been demonstrated in studies with IBD patients who displayed normal CRP values even though they had endoscopically active disease [6]. In the group of patients who do not show a CRP response over the course of the disease, SAA may represent a promising biomarker to be analysed simultaneously to CRP. Subsequently, the determination of SAA concentrations may help to determine treatment adjustments, to evaluate the need for endoscopy and to set the time interval for the next appointment.

This study has several limitations. Firstly, it is a single centre study with a retrospective design. Patients with missing laboratory markers and patients with an acute state of infection had to be excluded. Secondly, due to the retrospective character of the study, there were no results of endoscopic examinations, which could have been used to enhance the significance of the correlation and the cut-off value for SAA. The aim of future prospective studies should be the assessment of all laboratory parameters on the same day, as well as the performance of an endoscopic examination on that day, with larger cohorts of both disease entities. An inclusion criterion could be a normal CRP concentration whilst showing endoscopic activity. In addition, the assessment of SAA during a treatment course could be an interesting aspect for future studies. The missing documentation of the body mass index (BMI) is another limitation, as obesity can cause an increase in inflammatory biomarkers. Additionally, in our study, only one sample was included for the majority of the patients. In future studies, the SAA concentrations could be determined at different stages of the disease course, and in specified time intervals. This would also create the opportunity to assess the changes in SAA in response to therapy. Referring to laboratory methods, the determination of FC concentrations was limited due to the range of the central laboratory of the University Hospital Heidelberg. Exact values below 30 µg/g or above 1800 µg/g could not be determined.

Conversely, the strengths of the study include the large number of patients, with more than 100 patients per disease entity. Although SAA was already assessed in previous studies, the case number was often smaller and only included one disease entity. In addition, the exclusive use of biomarkers leads to a higher comparability of the samples and the correlation coefficients because their measurement is not subjective. Endoscopic scores can be difficult to compare and calculate in the first place because of the dependency on the clinician’s view. Finally, most of the studies did not include the correlation of faecal calprotectin and SAA in their analyses.

## 4. Materials and Methods

### 4.1. Patients

This is a retrospective uncontrolled monocentric study performed at the IBD outpatient clinic of the University Hospital Heidelberg. It serves as a tertial referral centre for IBD patients. The patients’ data were compiled from the clinic information system (ISH). The clinical data and serum samples were collected between April 2016 and November 2022 at the IBD outpatient clinic as part of the Department for Gastroenterology and Hepatology. For study purposes, the serum specimens were stored frozen at −80 °C if the patient gave written informed consent to use them for future studies. In total, there were 736 registered patients. For some of them, several serum samples from different time points were available.

Inclusion criteria for the patients were: age ≥ 18 years, diagnosis of IBD according to ECCO criteria [25,26], availability of at least one serum sample with a valid FC concentration measurement performed within 14 days before and 14 days after retrieval of the serum sample and plasma CRP measurement on the day of serum sample acquisition, no ileoanal pouch or ostomy and no obvious acute infection at the time of sample acquisition. Exclusion criteria were: age < 18 years, missing diagnosis of IBD, absence of a serum sample for SAA concentration measurement, absence of a valid FC concentration measurement and plasma CRP measurement, an ileoanal pouch or ostomy and the presence of an obvious acute infection at the time of serum sample acquisition.

### 4.2. Measurement of SAA and CRP Concentrations

For the measurement of SAA and CRP concentrations, serum and plasma from the day of the clinical examination were used. CRP was routinely measured in clinical practice and available in the laboratory record, while SAA was retrospectively analysed with routine diagnostic methods for the purpose of the study at the central laboratory of the University Hospital Heidelberg. The CRP concentrations were measured with an immune-turbidimetric assay in heparin plasma using a CRP-specific antibody. The analytic device was ADVIA XPT (Siemens Healthineers, Erlangen, Germany). SAA was determined from a serum sample by nephelometry using specific antibodies against human SAA. The device for this analysis was called BNII Siemens Healthcare Diagnostics (Siemens Healthineers, Erlangen, Germany). The cut-off value for active inflammation was set at 5 mg/L for CRP and at 6.4 mg/L for SAA concentration. For the lower limit of SAA concentrations (<3.5 mg/L) and CRP concentrations (<2.0 mg/L), the values 3.4 mg/L and 1.5 mg/L were chosen.

### 4.3. Measurement of FC Concentrations

The values for FC concentrations were available in the routine laboratory records within ±14 days of the date of the acquisition of the serum sample used to determine SAA. They were measured in stool samples at the central laboratory of the hospital with a quantitative enzyme-linked immunosorbent assay (ELISA). A photometer was used for this analysis. The cut-off value for healthy individuals is set at <50 µg/g. In conformity with previous studies, the cut-off value of the specific assay should be used for statistical analysis because there is still no standard cut-off for FC [27]. In the study, FC concentrations of <30 µg/g were set at 20 µg/g. The upper limit of the FC assay was 1800 µg/g. In the study, all concentrations of >1800 µg/g were set at 1900 µg/g.

### 4.4. Assessment of Clinical Disease Activity

To assess the clinical disease activity, the Harvey-Bradshaw Index (HBI) was used for CD patients [28], and the Simple Clinical Colitis Activity Index (SCCAI) was used for UC patients [29]. For patients classified as having IBDU, one of the two scores was applied, depending on the decision of the physician in charge of the clinical examination. Even if no clinical index was found in the record, patients were eligible to be included in the study, as the main goal was the correlation between laboratory markers.

### 4.5. Montreal Classification

The Montreal classification [30] was used to characterise disease behaviour in CD and UC (see Table 3 and Table 4).

### 4.6. Statistical Analysis

The statistical analysis was carried out with the program SPSS 2019 (version 29.0.0.0 (241)) for iOS (IBM Deutschland GmbH, Ehningen, Germany). For demographic variables, a presentation with means, standard deviations, medians and quartiles was chosen. The concentrations of biomarkers were expressed as medians with interquartile ranges. To test the data for normal distribution, a Q–Q Plot was used. Continuous variables were compared with the Mann–Whitney U Test, while the Chi-squared test was used for categorical variables. To analyse the correlation between two variables, the Spearman rank correlation coefficient was used. For all these tests a *p*-value of <0.05 indicates statistical significance. In addition, a receiver operating characteristic (ROC) analysis was performed, including the area under the curve (AUC). The Youden index was calculated in Microsoft Excel (Version 16.79) with the values from the ROC curve determining an optimal cut-off value for the serum SAA concentration used to detect active inflammation.

## 5. Conclusions

In conclusion, this study confirmed that an increase in SAA is associated with higher disease activity in patients with IBD. In addition, it may help to identify patients with active disease despite having normal CRP values. Therefore, it may be used as an additional biomarker to CRP and FC concentrations to guide treatment decisions in IBD patients. To confirm the benefit of SAA in this context, prospective studies should be performed where the usefulness of SAA concentrations to detect a response to IBD treatment are evaluated.

## Figures and Tables

**Figure 1 ijms-25-01177-f001:**
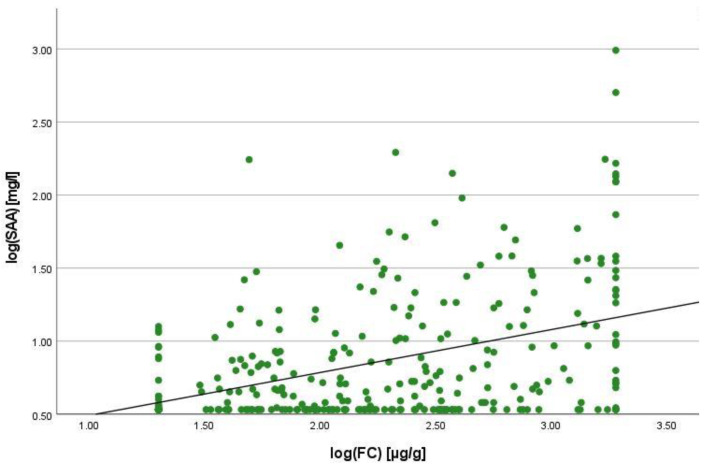
Scatterplot for log(SAA) and log(FC), SAA Serum Amyloid A, FC faecal calprotectin.

**Figure 2 ijms-25-01177-f002:**
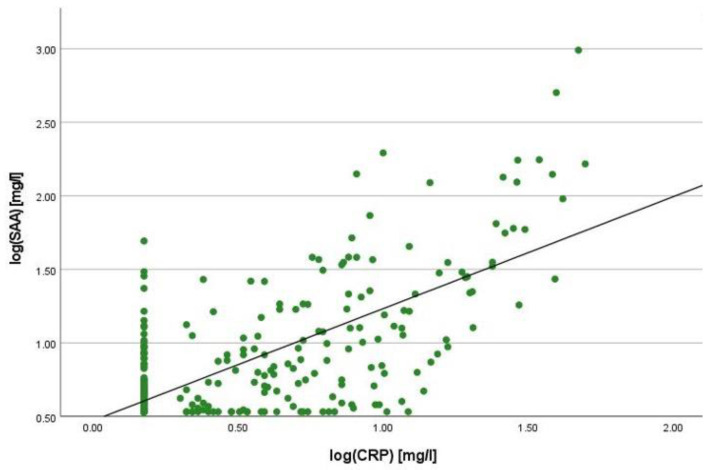
Scatterplot for log(SAA) and log(CRP), SAA Serum Amyloid A, CRP C-reactive Protein.

**Figure 3 ijms-25-01177-f003:**
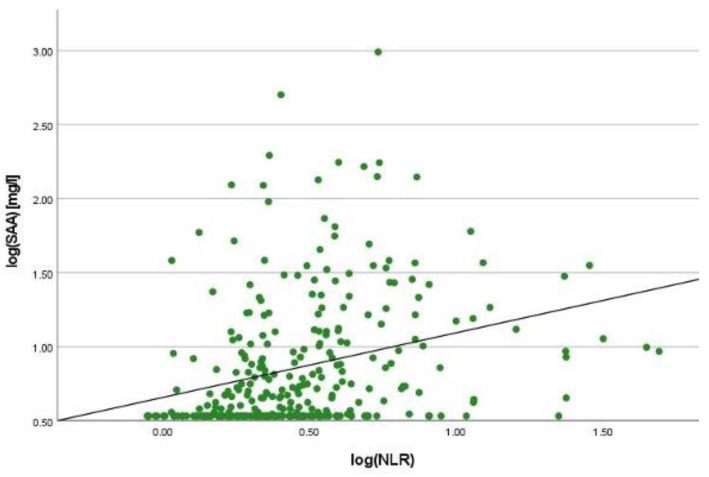
Scatterplot for log(SAA) and log(NLR), SAA Serum Amyloid A, NLR Neutrophil-to-lymphocyte-ratio.

**Figure 4 ijms-25-01177-f004:**
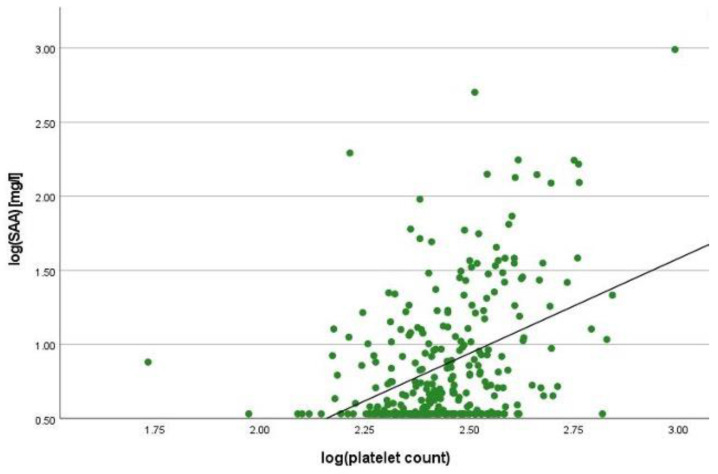
Scatterplot for log(SAA) and log(platelet count), SAA Serum Amyloid A.

**Figure 5 ijms-25-01177-f005:**
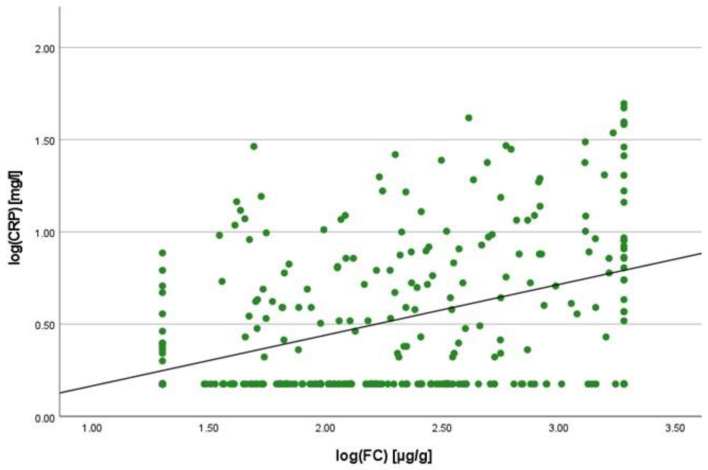
Scatterplot for log(CRP) and log(FC), CRP C-reactive protein, FC faecal calprotectin.

**Figure 6 ijms-25-01177-f006:**
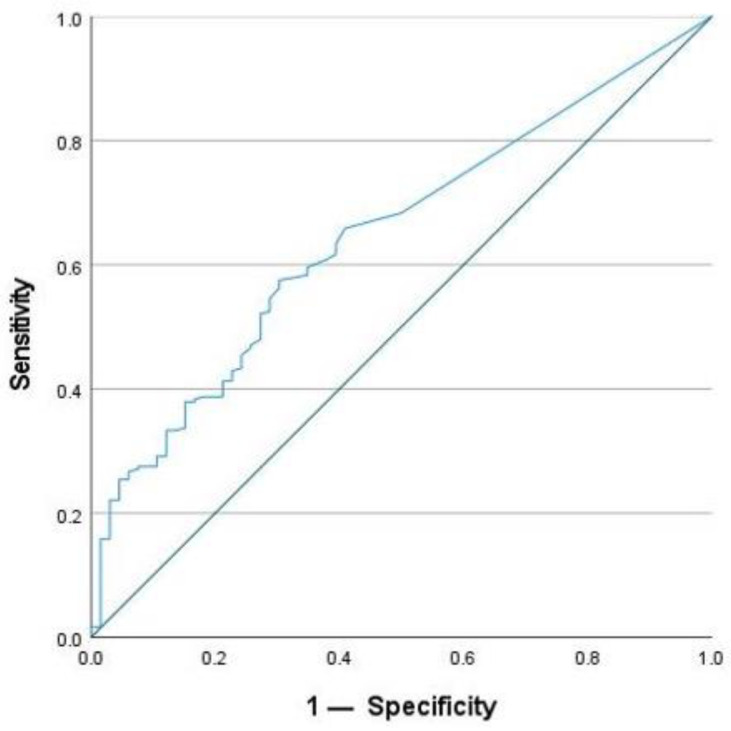
ROC-analysis for SAA to detect active inflammation.

**Figure 7 ijms-25-01177-f007:**
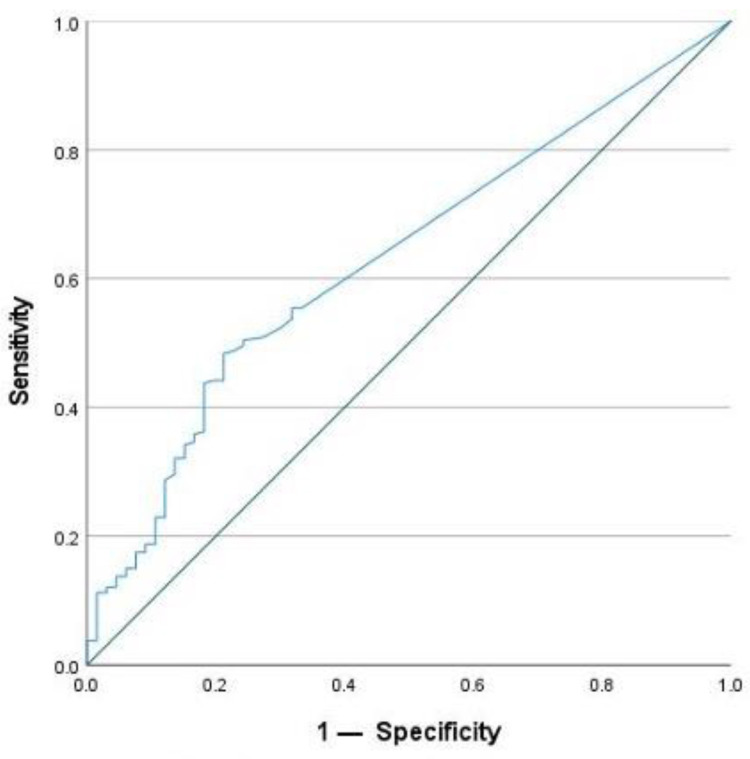
ROC-analysis for CRP to detect active inflammation.

**Figure 8 ijms-25-01177-f008:**
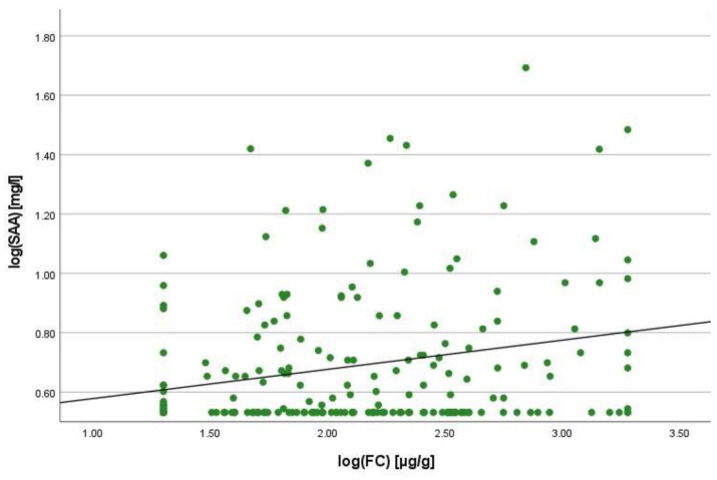
Scatterplot for log(SAA) and log(FC) for the group CRP <5 mg/L, SAA Serum Amyloid A, FC faecal calprotectin.

**Table 1 ijms-25-01177-t001:** Baseline characteristics of the included patients.

Parameter	Cases (306)
Age (years), median (range)	44 (18–77)
Female, *n* (%)	159 (52.0)
Male, *n* (%)	147 (48.0)
Crohn’s disease, *n* (%)	182 (59.5)
Ulcerative Colitis, *n* (%)	116 (37.9)
IBD unclassified, *n* (%)	8 (2.61)
Disease duration at serum sample collection (years), median (IQR)	10 (16)
Montreal Classification *n* (%)	
Age at diagnosis (MC and UC) (A1, A2, A3)	25 (8.2), 212 (69.3), 69 (22.5)
Location (MC) (L1, L2, L3, L4)	68 (37.4), 21 (11.5), 83 (45.6), 11 (6.0)
Behaviour (MC) (B1, B2, B3)	69 (37.9), 64 (35.2), 63 (34.6)
Location (CU) (E1, E2, E3)	12 (10.3), 44 (37.9), 64 (55.2)
History of resecting surgery for IBD, *n* (%)	85 (27.8)
Extraintestinal manifestation(s) (EIM), *n* (%) *	102 (33.3)
EIM: Peripheral arthritis, *n* (%)	81 (26.5)
EIM: Axial arthritis, *n* (%)	11 (3.59)
EIM: Skin, *n* (%)	16 (5.23)
EIM: Eyes, *n* (%)	7 (2.29)
EIM: Primary sclerosing cholangitis, *n* (%)	3 (0.98)
HBI/SCCAI (Mean) (SD)	3.2 (3.6)/2.9 (3.0)
Biologic therapy at the time of sample collection, *n* (%)	152 (49.7)
Number of days between FC and SAA determination, mean (SD)	3.7 (4.4)

HBI: Harvey Bradshaw Index, SCCAI: Simple Clinical Colitis Activity Index. * This is the number of cases with EIM. As some cases had more than one EIM, the sum of all subgroups is higher than this.

**Table 2 ijms-25-01177-t002:** Laboratory markers.

	*N*	Mean	Standard Deviation	Minimum	Maximum	Percentile		
						25	50 (Median)	75
FC concentration [µg/g]	306	446.1	591.1	20.0	1900.0	54.5	170.5	529.8
SAA concentration [mg/L]	306	18.7	67.6	3.40	980.0	3.40	4.70	10.9
CRP concentration [mg/L]	306	5.69	7.98	1.50	49.7	1.50	2.10	6.40
NLR	303	4.07	5.33	0.88	49.2	1.89	2.70	3.95
Platelet count	306	290.1	103.9	54.0	981.0	222.2	266.5	335.3

**Table 3 ijms-25-01177-t003:** Montreal classification for Crohn’s disease [30].

Age at diagnosis	A1: <17 years A2: 17–40 years A3: >40 years
Location	L1: ileal L2: colonic L3: ileocolonic L4: isolated upper disease
Behaviour	B1: non-stricturing, non-penetrating B2: stricturing B3: penetrating

**Table 4 ijms-25-01177-t004:** Montreal classification of extent of Ulcerative Colitis (UC) [30].

Extent	E1: Ulcerative proctitis E2: Left sided UC (distal UC) E3: Extensive UC (pancolitis)

## Data Availability

The data presented in this study are available on request from the corresponding author. The data are not publicly available due to data protection.

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
