# Peer review of "Serum Amyloid A as a Potential Biomarker in Inflammatory Bowel Diseases, Especially in Patients with Low C-Reactive Protein"

_ijms, 2024, doi:10.3390/ijms25021177_

Round 1

Reviewer 1 Report

Comments and Suggestions for Authors

In this retrospective study authors evaluated the clinical potential of SAA protein as a biomarker for IBD severity. The study group included 306 cases of IBD and no healthy controls. Obtained results do not indicate that SAA is a strong biomarker of disease severity that could replace existing markers. In the most optimistic scenario SAA could compete with serum CRP although the superiority of SAA is not evident. As pointed by the authors both markers are induced during various inflammatory processes hence their selectivity is rather poor. I would also like to mention that of all studied markers SAA had the highest SD which was approx. 3.6 times higher than the mean what indicates the high variability of this biomarker in the studied population. This could potentially be caused by another fact that authors mentioned in the introduction section, that SAA is an acute phase protein bound to high density lipoproteins (HDL) whose levels can increase 1000-fold during phases of acute inflammation. Moreover, as mentioned by the authors, previous studies already found a correlation between SAA concentrations and inflammatory markers such as CRP, FC, and interleukins, as well as clinical disease activity and endoscopic findings therefore this study adds very little to the scientific discourse. In addition, it is not clear what was the number of patients with normal SAA concentration and FC > 50 ug/g? It seems that it was quite high what suggests the rather low quality of SAA as a biomarker in IBD. Given all the issues above I do not recommend this study for publication.

Reviewer 2 Report

Comments and Suggestions for Authors

The article is interesting because it concerns an attempt to analyze the use of biomarkers such as C-reactive protein (CRP), fecal calprotectin (FC) and serum amyloid A (SAA) in inflammatory bowel diseases (IBD).   To detect phases of  clinical symptoms: active mucositis, disease activity should be monitored regularly. Sometimes a new exacerbation of the disease begins before the treatment is completed and the patient does not recognize any clinical symptoms. By far the most popular tool used to track colitis is endoscopy in combination with observed clinical symptoms and biomarkers in blood or stool. However, there are some disadvantages, for example endoscopy is invasive, time-consuming and, above all, a significant burden for the patient. Therefore, many studies are investigating non-invasive biomarkers for disease activity surveillance in both CD and ulcerative colitis.

I have some questions as follows:

1. What method and on what analyzer were the biomarkers determined: C-reactive protein (CRP), fecal calprotectin (FC) and serum amyloid A (SAA).

2. What potential pitfalls did the authors notice during stool sample collection and preliminary analysis?

3.Why are stool samples with calprotectin concentration > 1800 µg/g classified as 1900 µg/g?

4. In how many people have calprotectin levels above 1800 µg/g been observed?

5. Why were stool samples not diluted to obtain total stool FC concentrations?

6. Did the patients have other diseases, e.g. diabetes, obesity?

7. How many of the studied patient population were subjected to biological treatment?

8. How many patients received infliximab, adalimumab, golimumab, vedolizumab, ustekinumab, tofacitinib, and calcineurin  inhibitors)?

Reviewer 3 Report

Comments and Suggestions for Authors

The original article aimed to investigate the disease activity marker SAA in IBD patients concerning conventional CRP, DAIs, FC, and NLR.

Blood and stool samples from IBD patient groups (CD, UC, and UIBD) and normal control patients were used. The number of patients included is sufficiently large for a retrospective monocentric study.

The study methods and statistical procedures used are adequate.

It was found that there is a significant positive correlation between SAA and FC, CRP, NLR, and  DAIs. The cut-off for SAA serum concentration at 4.55 mg/l achieved a sensitivity of 57.5% and a specificity of 69.7% for the detection of active inflammation in IBD.

This publication concludes that SAA alone is not sufficient for the safe monitoring of IBD activity, but that it has clinical utility when used in combination with other markers to establish an inflammatory marker profile.

The study is well-designed; the aim is clear, and the study methods are appropriate.

However, I have a few comments on the results for consideration:

Treatment with either low doses (4 mg/day) of methylprednisolone or (5 mg/day) of prednisolone increases NLR, as steroids mobilize the marginal neutrophil pool. I think that for patients taking steroids, NLR alone or in combination is not a good choice to determine disease activity.

In addition to SAA, FC, NLR, CRP, and DAIs, I would also recommend the testing of other laboratory parameters that may also act as biomarkers of the acute phase. These are ferritin, platelet count, d-dimer, and IL-6. Why was SAA highlighted as the main subject of the study? Why?

Is SAA more specific for the intestine compared to SAA combined with SACE for sarcoidosis, which is more frequently and widely used in its diagnosis?

Is there a gut-specific marker at all?

There were patients with EIM included in the study.

What were the manifestations of EIM? Clarification and subgroup analysis of this question would be important, as the appearance of EIM, even in the absence of intestinal symptoms, is suggestive of an active IBD course. How did the assayed parameters change in the EIM group?

Did the SAA rate show a correlation with the ANCA profile of the patients and the ANCA titer? What was the ANCA profile observed in the studied patients? (ANCA titer, even in the case of x-ANCA positivity, may show a correlation with IBD activity.)

I think that answering the above questions and extending the studies is warranted. The article cannot be accepted for publication in its current form without addressing these questions and extending the studies.

Round 2

Reviewer 1 Report

Comments and Suggestions for Authors

As I pointed out in my previous review data presented by the authors do not add significantly to the existing knowledge and therefore is of low relevance to the scientific community. The amendments made in the manuscript did not change its general quality and therefore I uphold me previous decision and do not recommend this manuscript for publication.

Author Response

Dear reviewer, 

thank you for reading our manuscript again. We revised our manuscript according to the suggestions of the academic editor who referred to your comments. You can find the corresponding changes in the title and line 312 to 317, 327 to 328 and line 332 to 333. The changes are written in blue. 

Reviewer 3 Report

Comments and Suggestions for Authors

The authors have correctly made the proposed changes. They have answered the questions asked. What they did not answer, they explained why and included it in the article as an aspect to be considered in future research. 
I consider the corrected article acceptable for publication. 

Author Response

Dear reviewer, 

thank you for your time and effort to review the changes on our manuscript.